# Application of FBG Sensor to Safety Monitoring of Mine Shaft Lining Structure

**DOI:** 10.3390/s22134838

**Published:** 2022-06-26

**Authors:** Kun Hu, Zhishu Yao, Yanshuang Wu, Yongjie Xu, Xiaojian Wang, Chen Wang

**Affiliations:** 1School of Civil Engineering and Architecture, Anhui University of Science and Technology, Huainan 232001, China; kokonn0326@163.com (K.H.); wuyshuang@126.com (Y.W.); xyjyj666@163.com (Y.X.); xjwangyhl@163.com (X.W.); chenwecho@126.com (C.W.); 2Engineering Research Center of Underground Mine Engineering of Ministry of Education, Anhui University of Science and Technology, Huainan 232001, China

**Keywords:** mine shaft lining structure, vertical additional force, fiber Bragg grating (FBG) sensor, concrete strain, long-term safety monitoring

## Abstract

The use of fiber Bragg grating (FBG) sensors is proposed to solve the technical problem of poor sensor stability in the long-term safety monitoring of shaft lining structures. The auxiliary shaft of the Zhuxianzhuang coal mine was considered as the engineering background, and a test system implementing FBG sensors was established to monitor the long-term safety of the shaft lining structure. Indoor simulation testing revealed that the coefficient of determination (r^2^) between the test curves of the FBG sensor and the resistance strain gauge is greater than 0.99 in both the transverse and vertical strains. Therefore, the FBG sensor and resistance strain gauge test values are similar, and the error is small. The early warning value was obtained by calculation, according to the specific engineering geological conditions and shaft lining structure. The monitoring data obtained for the shaft lining at three test levels over more than three years reveal that the measured vertical strain value is less than the warning value, indicating that the shaft lining structure is currently in a safe state. The analysis of the monitoring data reveals that the vertical strain increment caused by the vertical additional force is approximately 0.0752 με/d. As the mine drainage progresses, the increasing vertical additional force acting on the shaft lining will compromise the safety of the shaft lining structure. Therefore, the monitoring must be enhanced to facilitate decision-making for safe shaft operation.

## 1. Introduction

To mine deep coal resources, a vertical access channel must be constructed from the ground surface to an underground space. This channel is called a vertical shaft, and it is an important channel for the lifting transport of coal (or rock refuse), personnel, materials, and equipment, in addition to functioning as ventilation and drainage during mine production [1,2]. The mine shaft is characterized by large depth, a large cross-sectional area, and complex hydro-geological conditions across strata [3]. The lining constructed to resist external loads, such as the soil and water pressure of a stratum, and maintain the stability of the vertical shaft is called the shaft lining structure [4,5].

When the vertical shaft passes through deep topsoil, it is necessary to adopt specific construction methods such as the artificial freezing method [6,7]. Artificial frozen ground drilling refers to the use of the artificial refrigeration method, whereby the water-bearing strata around the shaft become frozen soil, and a circular frozen wall is formed to pro-vide protection for shaft excavation and masonry construction. Finally, reinforced concrete is used to construct the shaft lining. When the shaft lining is appropriately constructed, the cold supply to the stratum stops, and the frozen soil wall begins to thaw until it completely melts. Subsequently, the reinforced-concrete shaft lining bears the water and soil pressure, self-weight, and shaft equipment weight to maintain shaft stability and operation safety [8]. Therefore, the shaft is the “throat” of the mine, and the shaft lining is the supporting structure that maintains the stability of the shaft. The shaft lining is designed to have a long service life, which is very important for ensuring the safety of coal mine production.

In the central and eastern mining areas of China, many coal mine shafts passing through different strata face particular stratum conditions; that is, the bottom aquifer of the topsoil is directly coated on the bedrock. Because there is no water insulation between the two layers, a hydraulic connection exists between them. The water level of the aquifer at the bottom of the topsoil decreases when mine drainage occurs. When the water level of the topsoil layer decreases, the pore water pressure also decreases, the effective stress increases, and formation consolidation settlement occurs. However, the vertical stiffness of the reinforced-concrete shaft lining located on the hard bedrock is large, and the compression deformation is small; therefore, the concrete shaft lining cannot sink synchronously with the stratum. When relative displacement is generated between the ground consolidation settlement and the reinforced-concrete shaft lining, an additional vertical force is exerted downward on the outer surface of the shaft lining and is similar to the negative friction force of pile foundation engineering [9]. Under the action of this force, the shaft lining produces considerable vertical compressive stress, which gradually increases from top to bottom and reaches the maximum value near the interface between the surface soil layer and the bedrock. With continuous mining, the stratum subsidence caused by the hydrophobic aquifer gradually increases, and the additional vertical force also increases. When the surface settlement reaches a certain value, the vertical compressive stress borne by the concrete of the shaft lining reaches the limit state, and the concrete of the shaft lining breaks (see Figure 1). In China, approximately 300 coal-mine shaft-lining-damage accidents have occurred, which poses a severe threat to the safety of mining operations [10,11,12].

To prevent the rupture of the shaft lining and ensure the safety of the mine, the evolution law of the additional vertical force on the shaft lining must be obtained. Because it is difficult to accurately calculate and predict the additional vertical force imposed on the shaft lining by using theoretical methods, owing to the existence of many influencing factors [13,14], the safety monitoring of the shaft lining stress is an effective technical method for preventing shaft lining damage.

By installing sensors on strategic points of a reinforced-concrete shaft lining structure, a monitoring system that can ensure continuous monitoring is established, and the stress and deformation of the structure can be monitored online in real time. Then the collected data are analyzed to predict the damage of the structure. Based on a set warning values, the safety state of the structure is analyzed over time. When the structure stress is close to the alarm threshold, an emergency plan is initiated to prevent the damage of the shaft lining structure [15].

Over the years, the long-term safety monitoring and accurate measurement of the stress change of the shaft lining structure during its operation have been perplexing problems for mine engineers and technicians. To monitor the safety of the shaft lining structure, resistance sensors or vibrating wire sensors have mainly been used as an effective method for monitoring the shaft lining stress and deformation in the short term [16,17]. However, in the complex mine-shaft environment, which is affected by multiple factors such as temperature, humidity, and electromagnetic interference, sensors have poor stability and can easily be corroded by moisture. Additionally, under electromagnetic interference, the test signal becomes unstable, and the test error increases. Consequently, it is difficult to use sensors in the long-term monitoring of shaft lining health, and this has become a major technical bottleneck in the monitoring of mine shaft safety.

In recent years, FBG sensors have been used in civil engineering as an effective means for monitoring the health of buildings. The FBG sensor uses an optical signal as the carrier of conversion and transmission and is characterized by a small size, light weight, high sensitivity, corrosion resistance, and strong anti-electromagnetic interference [18,19,20]. Particularly in humid environments, the FBG sensor has high test accuracy, good long-term stability, and no zero-shift, thus making it particularly suitable to the long-term safety monitoring of coal-mine shaft lining structures.

In past coal-mine shaft construction in China, substantial information-based construction monitoring was carried out for the shaft lining. For example, Song et al. [21] monitored the stress and deformation of the shaft lining during construction to provide a decision-making basis for ensuring the safe completion of the shaft lining. With regard to the additional vertical force, many studies have carried out stress and deformation monitoring for in-service shaft linings, analyzed the safety state of the shaft lining, and predicted the possible damage of the shaft lining [22,23,24,25]. However, when using resistance sensors or vibrating wire sensors, the monitoring time is relatively short; therefore, the long-term safety monitoring of the shaft lining is not possible. Wang et al. [26] used FBG sensors to monitor the stress during shaft drilling and analyzed the stress law of the shaft lining. However, few studies have used FBG sensors to monitor the safety of coal-mine shaft lining structures in the long term. Specifically, there is a lack of in-depth studies on measurement accuracy, installation methods, and sensor effectiveness in complex environments. To apply FBG sensors to the long-term monitoring of coal-mine shaft lining safety, such that the safety state of the shaft lining structure can be evaluated in real time and shaft lining damage can be prevented, this study investigated the application of FBG sensors to the safety monitoring of mine shaft lining structures.

## 2. Testing Principle of FBG Sensor

The fiber Bragg grating (FBG) sensor is a type of fiber grating strain that is formed by changing the refractive index of the fiber core to produce small periodic modulation [27,28]. When the temperature or stress conditions change, the fiber produces axial strain, which increases the grating period and decreases the radius of the fiber core and cladding. The refractive index of the fiber is changed by photoelastic effects, which shift the grating wavelength. By considering the linear relationship between the strain and the grating wavelength offset, the strain of the measured structure can be calculated [29,30].

As shown in Figure 2, the FBG strain sensor is formed by the periodic change of the refractive index of the fiber core along the axial direction of the fiber. When the incident laser wavelength and the period of the FBG satisfy the condition of Equation (1), the grating will reflect the laser. Equation (1) relates the reflected wavelength of FBG ($\lambda_{B}$) to the grid spacing and refractive index of the fiber. When the axial deformation and temperature change of the fiber cause the drift of the grid spacing and refractive index, the reflected wavelength shifts accordingly, and the deformation or temperature change of the fiber can be obtained by measuring the drift of $\lambda_{B}$ [31].
(1)$$\lambda_{B}=2n_{eff}\Lambda$$
where $\lambda_{B}$ is the central wavelength, $n_{eff}$ is the refractive index of the fiber, and Λ is the grid spacing.

Previous studies have reported that both the strain and temperature have a good linear relationship with the central wavelength ($\lambda_{B}$) and are independent of each other. Under ideal conditions, the fiber is only affected by stress or temperature changes, and the correlation formulas are expressed as follows:(2)$$\Delta \lambda_{Bs}=\alpha_{\varepsilon }\varepsilon \;$$
(3)$$\Delta \lambda_{Bt}=\alpha_{T}\Delta T$$
where $\Delta \lambda_{Bs}$ is the central wavelength shift that is only affected by stress, $\Delta \lambda_{Bt}$ is the central wavelength shift that is only affected by temperature, $\alpha_{\varepsilon }$ is the strain sensitivity coefficient of the fiber grating, $\alpha_{T}$ is the temperature sensitivity coefficient of the fiber grating, ΔT is the temperature change value, and ε is the strain. 

However, in actual long-term shaft lining health monitoring, the shaft lining concrete is affected by both the stress and temperature; therefore, the actual central wavelength drift, $\Delta \lambda_{B}$, is superimposed by $\Delta \lambda_{Bs}$ and $\Delta \lambda_{Bt}$, and the correlation formula is expressed as follows [32,33]:(4)$$\Delta \lambda_{B}=\alpha_{\varepsilon }\varepsilon +\alpha_{T}\Delta T$$

Presently, the wavelength demodulation accuracy of FBG can reach 1 pm, the corresponding strain measurement accuracy can reach 1 micro strain, and the temperature measurement accuracy is 0.1 °C [34].

Because the FBG sensor can accurately measure the micro deformation of structural materials, it can be used to develop various sensors by attaching FBG sensor to elastic components and packaging to measure the strain, displacement, pressure, and temperature.

## 3. Indoor Simulation Test of Shaft Lining Safety Monitoring Sensor

### 3.1. Test Contents and Methods

Owing to the immaturity of FBG sensors with regard to monitoring the long-term safety of mine shaft lining structures, the test accuracy of the FBG sensor and that of the resistance strain gauge commonly used in the laboratory were compared through indoor simulation testing.

Before the test, a short concrete column with the size of 300 × 300 × 1000 mm was fabricated; the concrete strength grade was C50. Then FBG sensors were installed on the front and rear sides of the concrete column, as shown in Figure 3. The main parameters are listed in Table 1. Moreover, the resistance strain gauge was attached to the left and right sides of the concrete column.

The FBG strain sensor collected data by using the NZS-FBG-A04 multichannel fiber grating demodulator, as shown in Figure 4.

A particular type of strain gauge for rubber base concrete with a distance of 50 mm was used as the resistance strain gauge, and a YE253 static resistance strain gauge was used for data acquisition.

The support installation method was adopted for the FBG strain sensor; that is, holes were drilled on the surface of the concrete column, and the support was fixed. Two sensors were installed on each surface in the horizontal and vertical direction, respectively. The resistance strain gauge was attached to the center of both sides of the specimen, using 502 glue, with one piece attached horizontally and the other piece attached vertically, and connected to the strain gauge through a test wire. The detailed installation is shown in Figure 5 and Figure 6.

After the sensor components and measuring instruments were installed and pre-loaded, the readings of each instrument were checked to ensure that they were normal. The formal test started after the check was completed and the readings were confirmed.

### 3.2. Experimental Results and Analysis

The CSS-YAW3000 long-column electro-hydraulic servo pressure testing machine was used in the loading test of the concrete specimens. The method of graded loading was adopted. Each time, 100 kN was applied, and the maximum loading was 1500 kN. The test was repeated three times, and the average value of the three measurements was taken, as shown in Figure 7 and Figure 8.

As shown in Figure 7, in the loading process, when the load increased by 100 kN, the transverse strain of the resistance strain gauge increased by approximately 7.2 με, and the transverse strain of the FBG sensor increased by approximately 7.86 με. When loaded to 1500 KN, the maximum transverse strain of the resistance strain gauge and FBG sensor were 115 με and 120.02 με, respectively. As shown in Figure 8, in the vertical direction, when the load increased by 100 kN, the strain of the resistance strain gauge increased by approximately 32.5 με, and the strain of the FBG sensor increased by approximately 31.79 με. When loaded to 1500 kN, the maximum vertical strains of the resistance strain gauge and FBG sensor were −490 με and −470.82 με, respectively, which is essentially the same. Additionally, the correlation between the strain curves of the FBG sensor and the resistance strain gauge is calculated by using the CORREL function, and the coefficient of determination (r^2^) of the two curves under transverse strain is 0.9982. In the case of vertical strain, the coefficient of determination (r^2^) of the two curves is 0.9985. Therefore, regardless of the deformation being transverse or vertical, the FBG sensor and resistance strain gauge test values were very close, and the error was miniscule, thus indicating that the use of the FBG sensor in the concrete structure deformation test is feasible and the test accuracy is sufficiently accurate. Compared with the conventional strain sensor, the FBG sensor has remarkable anti-interference ability and can adapt to complex environmental conditions. Therefore, the FBG sensor is ideal for monitoring the safety of mine shaft lining structures.

## 4. Long-Term Safety Monitoring of Mine Shaft Lining Structure

### 4.1. Engineering Situations

The Zhuxianzhuang coal mine is located in Huaibei Plain, China. There are three vertical shafts in the industrial square, namely the main shaft, auxiliary shaft, and air shaft. Among them, the design net diameter of the auxiliary shaft is 6.0 m, and the shaft depth is 500 m. The shaft passes through the topsoil layer with a thickness of 254 m, which consists of four water-bearing layers and three water-resisting layers. The bottom water-bearing layer lies directly on the bedrock and belongs to a stratum with particular hydrophobic conditions. Because the auxiliary shaft passes through the deep topsoil, the shaft construction adopts the artificial freezing method, and the support form is the reinforced concrete shaft lining structure. The strength grade of the designed concrete is C30, and the maximum thickness of the shaft lining is 1200 mm.

Because the bottom water-bearing layer of the topsoil (referred to as the fourth aquifer) lies directly on a coal stratum, the drainage of mine excavation will inevitably de-crease the water level of the fourth aquifer. As the water level in the fourth aquifer de-creases, the effective stress increases, and the stratum consolidates and settles; this, in turn, exerts a considerable additional vertical force on the reinforced-concrete shaft lining. When this force exceeds the ultimate bearing capacity of the shaft lining, the shaft lining will rupture, and this will severely compromise the safety of mine operations. Therefore, the long-term monitoring of the safety of the shaft lining structure has great engineering significance.

### 4.2. Formulation of Monitoring Scheme

The coal mine shaft is a narrow cylindrical space located below the surface, and it is subject to high air humidity, occasional showers, the laying of communication and power cables, electromagnetic interference from many sources, and poor environmental conditions. Therefore, the FBG sensor with excellent performance was selected to monitor the auxiliary shaft lining of the Zhuxianzhuang coal mine.

#### 4.2.1. Design of Monitoring Level

According to engineering geological data and the shaft lining structure diagram, three monitoring levels were designed and located in the middle of the thick clay layer in the third water-resisting layer and the middle and bottom of the fourth water-bearing layer (bottom water-bearing layer), respectively. The specific test level design is presented in Table 2.

#### 4.2.2. Layout of Monitoring Elements

At each monitoring level, four measuring points were arranged equidistantly on the inner surface of the shaft lining, located in the east, south, west, and north directions, respectively. One vertical and one circumferential FBG strain sensor were arranged at each measuring point, and eight FBG sensors were arranged at each monitoring level. The specific layout is shown in Figure 9 and Figure 10. The established monitoring system is shown in Figure 11.

#### 4.2.3. Determination of Early Warning Value of Shaft Lining Safety Monitoring

Based on the code for design of coal mine shaft and chamber (GB50384-2007) [35] and the code for design of concrete structures (GB50010-2010) [36], and according to the engineering geological conditions and the shaft lining structure of the corresponding layers of the auxiliary well monitoring level, the early warning values of the different levels of the shaft lining safety monitoring were obtained through relevant calculations, as presented in Table 3.

### 4.3. Monitoring Results and Analysis

For the long-term monitoring of the safety of the auxiliary shaft lining in the Zhuxian-zhuang coal mine, the FBG sensor installation began in early July 2018, and the monitoring system was established by the end of August 2018. Subsequently, real-time monitoring was carried out, and a large amount of test data was obtained. The main test results obtained by analysis and processing are shown in Figure 12, Figure 13 and Figure 14.

As shown in Figure 12, after the installation of the test elements, the maximum vertical tensile strain of the shaft lining concrete at the first test level was obtained as 28.73 με, the maximum vertical compressive strain was obtained as −220.53 με, and the vertical strain was found to be low. As presented in Table 3, the measured vertical strain of the shaft lining concrete is far less than the warning value, thus indicating that the shaft lining structure at this position is currently in a safe state.

As shown in Figure 13, after the formation of the test system, the measured maxi-mum vertical tensile strain of the shaft lining concrete at the second test level was 20.52 με, and the maximum vertical compressive strain was −206.48 με. As presented in Table 3, the measured vertical strain in the shaft lining structure is currently less than the warning value, meaning that the shaft lining structure is in a safe state at the second test level.

As shown in Figure 14, after the establishment of the test system and the beginning of the test, the maximum vertical tensile strain of the shaft lining concrete at the third test level was obtained as 18.81 με, and the maximum vertical compressive strain was obtained as −179.30 με. As presented in Table 3, the vertical strain of the shaft lining concrete measured at this position is less than the warning value, thus indicating that the shaft lining structure is in a safe state at the third test level.

The variation law of the vertical strain of the shaft lining concrete was analyzed based on the vertical strain of the shaft lining measured from September 2018 to December 2021. Figure 12 shows the vertical strain curve obtained by the first monitoring level of the auxiliary shaft lining in the Zhuxianzhuang coal mine. As can be seen, the vertical strain of the shaft lining concrete periodically fluctuates with the seasons in a certain range, thus indicating the close correlation between the vertical strain of the shaft lining and the seasonal change of temperature. In the Huaibei area, wherein the Zhuxianzhuang coal mine is located, the temperature difference between winter and summer is approximately 25 °C. Under the structural constraints of the shaft lining concrete, the analysis of test data reveals that the vertical strain fluctuation value of the shaft lining concrete caused by temperature difference is approximately 160 με. The difference between the maximum and minimum vertical strains measured at this level is 249.26 με. Therefore, the strain caused by the vertical additional force is approximately 89.26 με, and in the current monitoring period, the vertical strain increment caused by the vertical additional force is approximately 0.0752 με/d. With continuous mining, the bottom water-bearing layer of the topsoil continues to decline, and the soil layer consolidates and subsides. Hence, the vertical additional force acting on the shaft lining will continue to increase, and this is detrimental to the safety of the shaft lining structure. Therefore, enhanced monitoring and prevention measures are required. Although additional strain accumulates and slows down alternately with the seasons, owing to seasonal temperature changes, shaft lining damage can easily occur owing to the large value of seasonal strain in summer and its superposition with the vertical strain induced by the additional vertical force. This is also the root cause of most shaft lining damage that occurs before summer, and attention is required in this regard.

Under the normal operation of the shaft lining structure, the vertical strain change of concrete results from the combined action of the temperature change and additional vertical force. The curve of the vertical strain caused by the periodic fluctuation of temperature exhibits similar fluctuation, and the vertical strain caused by the additional vertical force exerted by formation settlement gradually accumulates and plays a leading role in the long-term stress state of the shaft lining.

Notably, if only the strain generated by the vertical additional stress needs to be analyzed, an optical fiber thermometer can be added in the adjacent position of the FBG sensor to measure the strain generated by the temperature change alone. Then the in-fluence of temperature in the measured value of the FBG sensor can be removed to achieve temperature compensation for the FBG sensor [37].

## 5. Preplan for Preventing Shaft Lining Damage

Based on the data collected by the shaft lining safety monitoring system, the safety of the shaft lining can be evaluated in real time. When the monitoring result reaches the yellow alert value, the plan for the prevention of shaft lining damage is initiated, and construction preparations are made. The plan is immediately executed when the monitoring result is close to the orange alert value. The specific plan is described below.

Considering that the hydrophobic settlement of the topsoil layer of the auxiliary shaft in the Zhuxianzhuang coal mine may cause shaft lining damage, owing to the coupling mechanism of the strata and shaft lining, and based on relevant engineering practice, it is proposed that a pressure-relief groove is opened at the inner shaft lining to prevent shaft lining damage [38,39,40].

To reduce the additional vertical force acting on the shaft lining during drainage and ensure the safety of the shaft lining structure, a pressure-relief groove was constructed at the depth of 259.5 m, as shown in Figure 15.

The advantages and disadvantages of the pressure-relief groove scheme are as follows: there is an obvious release and attenuation of the vertical additional force acting on the shaft lining, the prevention of shaft lining rupture is satisfactory, and the engineering cost is low. However, the construction in the shaft lining needs to occupy the lifting time, the construction environment is not good, and the safety conditions are poor.

After monitoring for half a year, the strain values measured by sensors at each level of the auxiliary shaft lining in the Zhuxianzhuang coal mine are smaller than the early warning value, thus indicating that the shaft lining is still in a relatively safe and stable state. However, as the water discharge progresses, the strain of concrete, particularly the vertical strain, increases. Therefore, attention should be paid to shaft lining stability.

## 6. Conclusions

Owing to the drainage of mine excavation and surface subsidence, a considerable additional vertical force is exerted on the shaft lining, and this can easily lead to shaft lining rupture. To ensure the stable performance of sensors in the monitoring of mine shaft lining safety in the long term, a test system implementing FBG sensors is proposed. Considering the auxiliary shaft of Zhuxianzhuang coal mine as the engineering background, this study analyzed the mechanical deformation mechanism and conducted safety monitoring for the auxiliary shaft lining of the Zhuxianzhuang coal mine by combining indoor sensor performance testing and field monitoring. The main conclusions drawn from this study are as follows.

(1) Considering the complex environmental conditions in the auxiliary shaft of Zhuxianzhuang coal mine, such as humid air, dripping water, and large interference of optical and magnetic signals, a system implementing FBG sensors is proposed for the long-term safety monitoring of the mine shaft lining structure based on analysis and comparison with existing strain sensors.

(2) The indoor simulation test results reveal that the values measured by the FBG sensor and the resistance strain gauge under the same loading conditions are very close, the error is small, and the change trend is the same, thus indicating that the FBG sensor has high accuracy. Additionally, the FBG sensor has remarkable anti-interference ability compared with conventional sensors.

(3) The early warning value of concrete vertical strain for the long-term safety monitoring of the auxiliary shaft lining in the Zhuxianzhuang coal mine was obtained through the analysis and calculation of the shaft lining stress. The construction of a pressure-relief groove at the inner shaft lining is proposed to prevent damage to the shaft lining structure by additional vertical force accumulation. When the test results reach the yellow alert value, the construction plan is immediately initiated to prevent shaft lining damage.

(4) The long-term monitoring of the shaft lining in the Zhuxianzhuang coal mine at three test levels for more than three years shows that the maximum vertical tensile strain of concrete in the shaft lining structure is 28.73 με, and the maximum vertical compressive strain is −220.53 με. The measured vertical strain values are less than the warning value, thus indicating that the shaft lining structure is currently in a safe state.

(5) The analysis of monitoring data reveals that the vertical strain of the shaft lining concrete fluctuates periodically with the seasons in a certain range. In the auxiliary shaft of the Zhuxianzhuang coal mine, the vertical strain fluctuation of the shaft lining concrete caused by the temperature difference between winter and summer is approximately 160 με. Currently, the vertical strain increment caused by the additional vertical force is approximately 0.0752 με/d. As the drainage of mine excavation continues, the additional vertical force acting on the shaft lining continues to increase, and this is detrimental to the safety of the shaft lining structure. Therefore, the monitoring must be enhanced to prevent accidents.

This study demonstrated the applicability of the FBG sensor in the long-term monitoring of mine-shaft lining health and provides a basis for the popularization and application of the FBG sensor to the health monitoring of the mine shaft’s lining.

In future work, with the popularization of the FBG monitoring system in mine monitoring, the informatization degree of the monitoring system can also be strengthened on this basis. The FBG sensor can be endowed with more intelligent elements. For example, the intelligent FBG sensor can cooperate with a small number of base stations arranged in the mine to receive wireless signals and, thus, realize the wireless transmission of measurement data and more predictable shaft lining structural health monitoring through Big Data processing. This will not only reduce the cable layout of the monitoring system but will also make the operation of the entire system more efficient and intelligent.

In conclusion, the FBG sensor is suitable for wide application to the long-term health monitoring of mines and has good potential for further development.

## Figures and Tables

**Figure 1 sensors-22-04838-f001:**
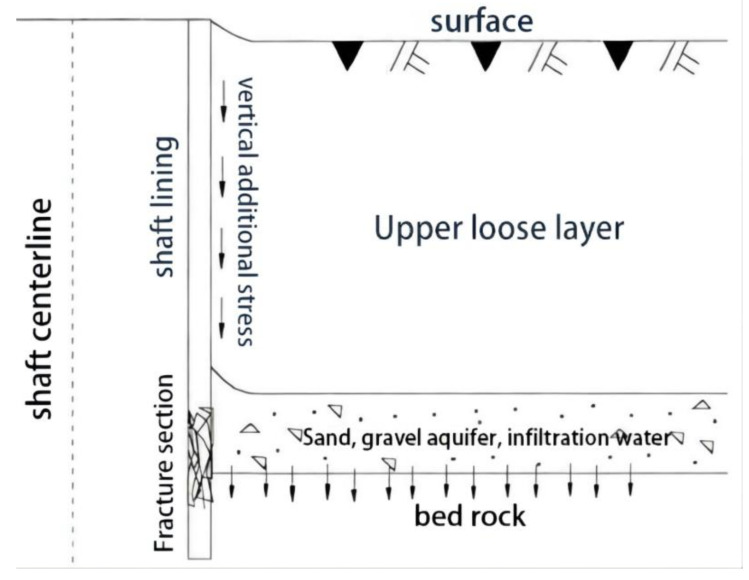
Schematic diagram of shaft lining fracture mechanism.

**Figure 2 sensors-22-04838-f002:**
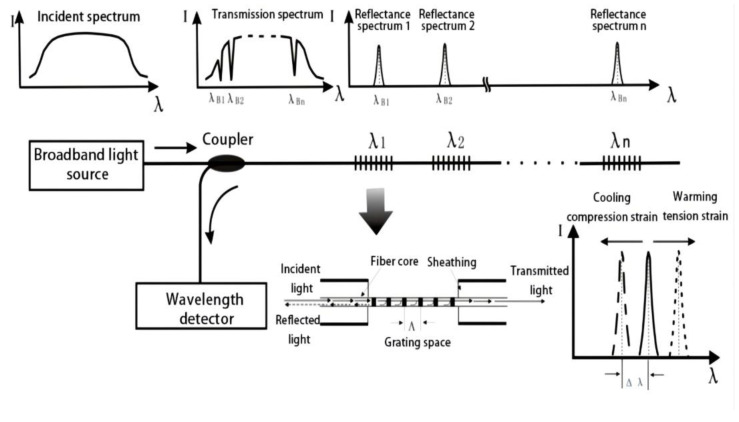
Testing principle of FBG strain sensor.

**Figure 3 sensors-22-04838-f003:**
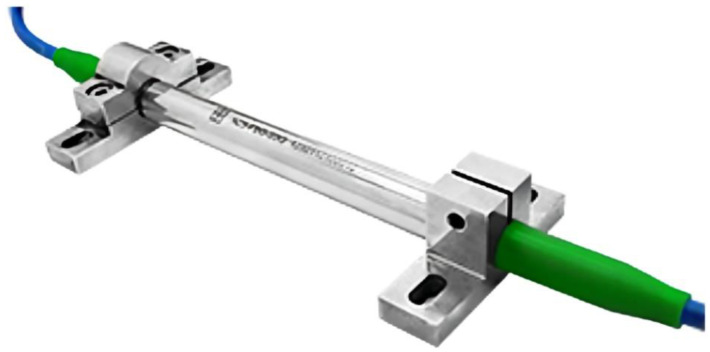
FBG strain sensor.

**Figure 4 sensors-22-04838-f004:**
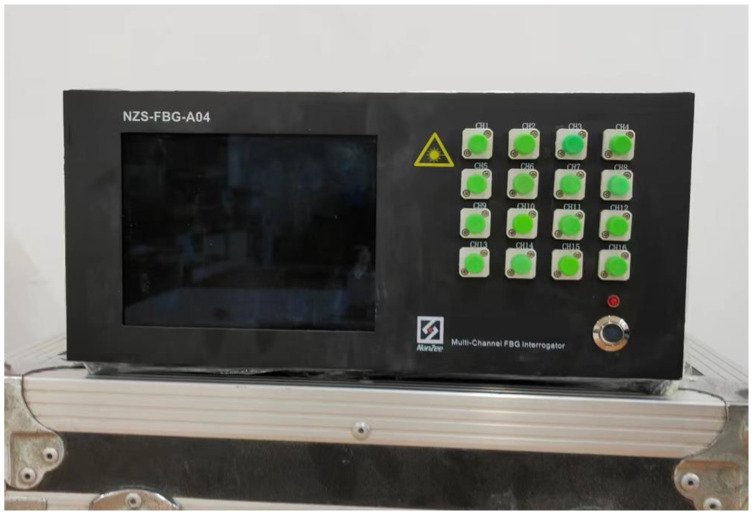
NZS-FBG-A04 multichannel fiber grating demodulator.

**Figure 5 sensors-22-04838-f005:**
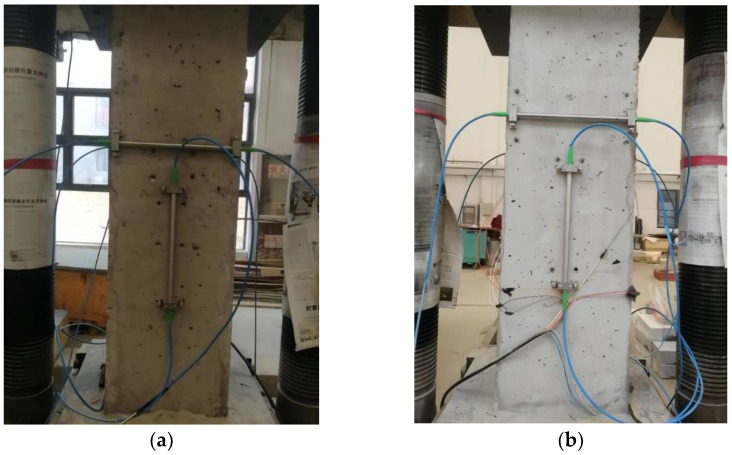
Installation diagram of FBG strain sensor: (**a**) front side and (**b**) rear side.

**Figure 6 sensors-22-04838-f006:**
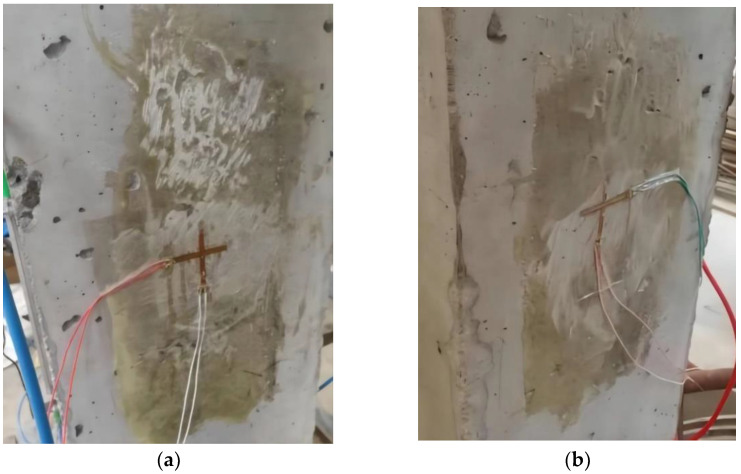
Installation diagram of resistance strain gauge: (**a**) left side and (**b**) right side.

**Figure 7 sensors-22-04838-f007:**
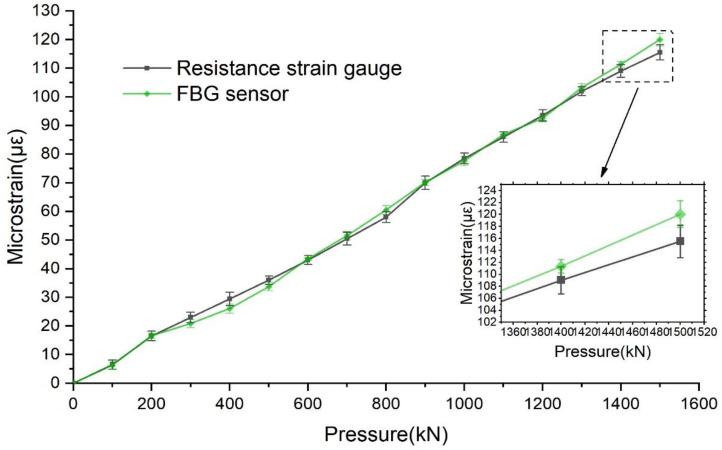
Comparison between curve of transverse strain of FBG sensor and resistance strain gauge.

**Figure 8 sensors-22-04838-f008:**
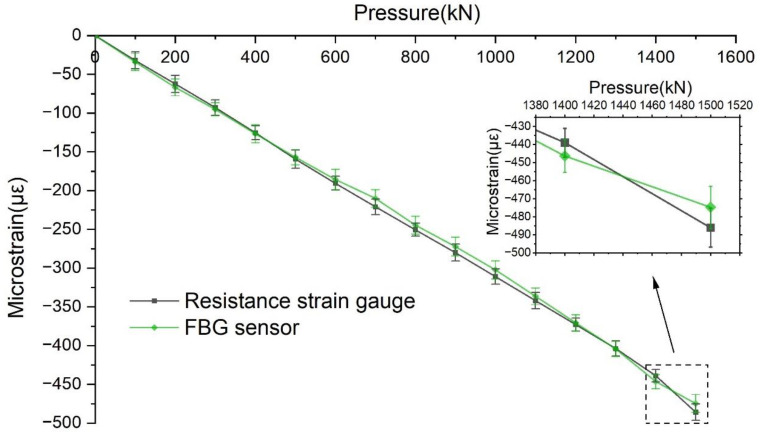
Comparison between curve of vertical strain of FBG sensor and resistance strain gauge.

**Figure 9 sensors-22-04838-f009:**
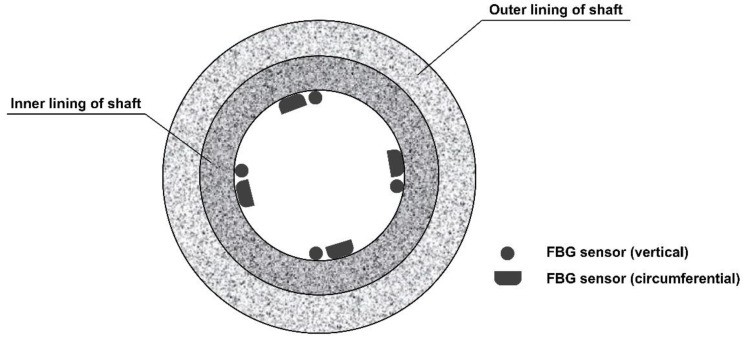
Schematic diagram of layout of FBG sensors.

**Figure 10 sensors-22-04838-f010:**
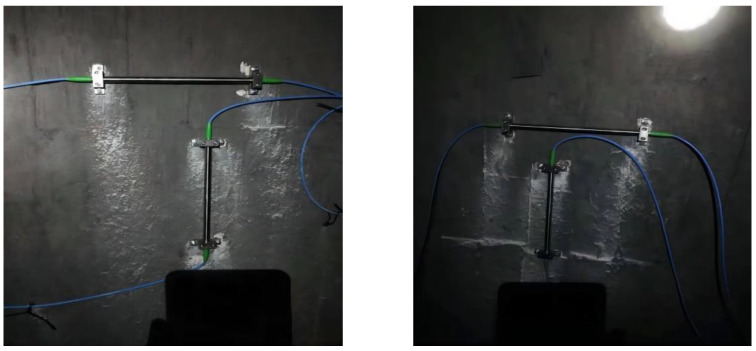
Field layout and installation of FBG sensors.

**Figure 11 sensors-22-04838-f011:**
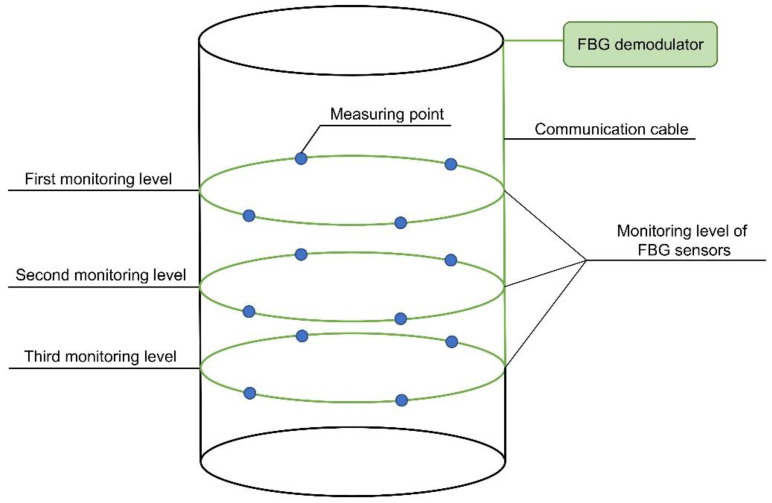
Schematic diagram of FBG monitoring system for shaft lining structure.

**Figure 12 sensors-22-04838-f012:**
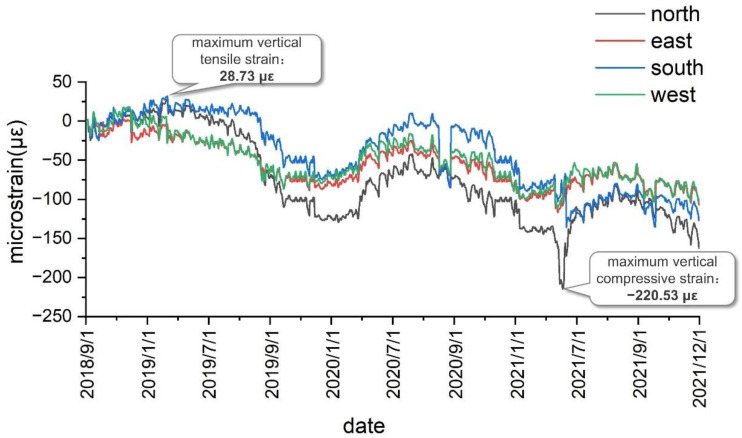
Measured vertical strain curves at first monitoring level.

**Figure 13 sensors-22-04838-f013:**
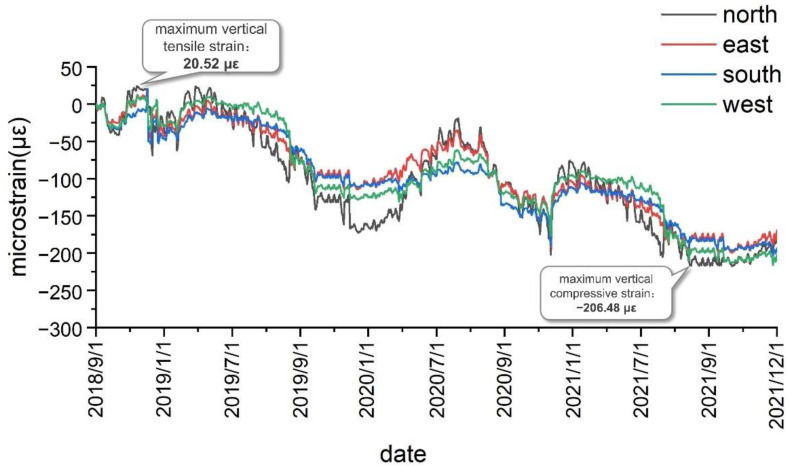
Measured vertical strain curves at second monitoring level.

**Figure 14 sensors-22-04838-f014:**
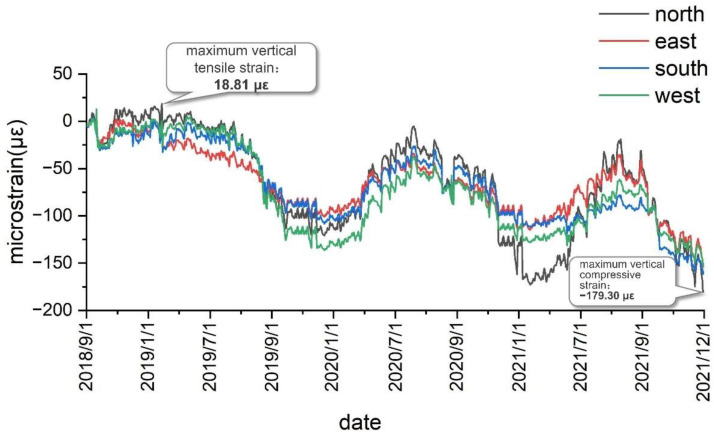
Measured vertical strain curves at third monitoring level.

**Figure 15 sensors-22-04838-f015:**
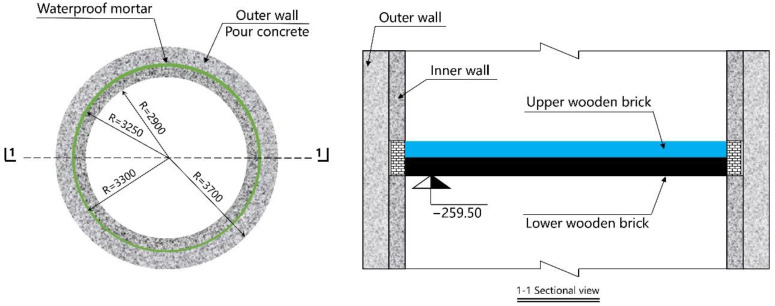
Schematic diagram of shaft-lining-damage prevention by cutting pressure-relief groove method.

**Table 1 sensors-22-04838-t001:** Main parameters of the FBG strain sensor.

Parameter Type	Parameter Value
Measuring range	−1500 to +1000
Precision	1‰ F.S.
Resolution	0.5‰ F.S.
Central wavelength of grating (nm)	1510–1590
Reflectivity	≥90%
Response time (s)	0.1
Size (mm)	Φ12 × 126
Connection mode	Fusion welding or FC/APC plug
Installation method	Welding or supporting installation

**Table 2 sensors-22-04838-t002:** Locations of monitoring level design.

Level Number	Level Depth/m	Corresponding Stratum	Soil Property
1	212	Middle part of third water-resisting layer	Clay
2	239	Middle part of fourth water-bearing layer	Sand
3	254	Bottom of fourth water-bearing layer	Gravel

**Table 3 sensors-22-04838-t003:** Early warning value of vertical strain (με) for shaft lining safety monitoring.

Monitoring Level	Force Situation	Yellow Alert	Orange Alert	Red Alert
1	Compression	−677.7	−690.4	−707.4
1	Tension	128.3	142.8	157.2
2	Compression	−663.0	−677.4	−696.6
2	Tension	139.2	153.6	168.0
3	Compression	−654.9	−670.2	−690.6
3	Tension	145.2	159.6	174.0

## Data Availability

Not applicable.

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
