# Peer review of "Application of FBG Sensor to Safety Monitoring of Mine Shaft Lining Structure"

_sensors, 2022, doi:10.3390/s22134838_

Round 1

Reviewer 1 Report

In this paper, the fiber grating sensor is applied to the safety detection of the mine shaft wall structure, and the stress sensing ability of the fiber grating sensor is used to detect the stress of the mine shaft wall, and the actual experimental verification is carried out. The experimental design of this paper is reasonable, the experimental data is sufficient, and it can be published directly after some modifications.

1)The full name should be added to the first occurrence of "FBG" in the abstract.

2)References should be added where Equation 1 and Equation 2 are and require further explanation.

Author Response

请参阅附件。

Reviewer 2 Report

The authors demonstrated a new method for Application of FBG sensor to safety monitoring of mine shaft lining structure. The measured vertical strain values are less than the warning value. The analysis of the monitoring data reveals that the vertical strain increment caused by the vertical additional force is approximately 0.0752 με/d. The paper makes a good contribution in terms of application in the field. Yet, some descriptions are not clear in the manuscript. I recommend publication, if the following mandatory revisions are made well. Below are some specific comments:

1.      Abstract: FBG acronym must be described.

2.      Abstract- [Page 1, line 14-15]: Indoor simulation testing revealed that the FBG sensor and resistance strain gauge test values are similar, and the error is small. Some details about the quantitative comparison results should be provided, rather than a simple statement (.... revealed that the FBG sensor and resistance strain gauge test values are similar, and the error is small.....).

3.      The author said in Introduction [Page 3, line 90]: in the complex mine shaft environment, which is affected by multiple factors such as temperature, humidity, and electromagnetic interference, sensors have poor stability and can easily be corroded by moisture. Please strengthen what kind of innovative and groundbreaking technological improvements have been made in this study using FBG sensors for long-term monitoring of coal mine shaft wall safety.

4.      [Page 4, line 141]-Experimental investigations have revealed that both the strain and temperature have a good linear relationship with the central wavelength and are in dependent of each other.  And [Page 4, line 146-148] ... the wavelength demodulation accuracy of FBG can reach 1pm, the corresponding strain measurement accuracy can reach 1 micro strain, and the temperature measurement accuracy is 0.1 °C. This should be an important result for the FBG sensor and it is recommended to add this data graph.

5.      [Page 7, line 194]-The test was repeated three times, and the average value of the three measurements was taken, as shown in Figures 7–8. It is recommended to add the standard deviation or y-error in Figures 7 and 8, and increase the value of the correlation linearity coefficient (r^2) for comparison, so that your results are more meaningful.

6.      [Page 8, line 245-249]-The author designed three monitoring layers, the layer depth/m is 212, 239, 254 m, respectively, the interval between each depth was about 15 m. Whether the author has tested the separation distance greater than 15m or less than 15m. Will the measurement result be the same?

7.      Figure 12, 13,14 has weak quality. (graphic overlapping is not easy to distinguish) Please improve. In addition to this, the authors are advised to add more detailed explanatory text or labels to the maximum vertical tensile strain and the maximum vertical compressive strain, and the reader will be able to understand these figures.

8.      [Page 9, line 253-257] - At each monitoring level, four measuring points were arranged equidistantly on the inner surface of the shaft lining, and located in the east, south, west, and north directions, respectively. Figure 12-14 Although the measurement trends are the same in the east, west, south and north, why does the signal at the measurement point in the north drop the most and has the largest jitter rate?

9.      [Page 12, line 311-325] - Under the structural constraints of the shaft lining concrete, the analysis of test data reveals that the vertical strain fluctuation value of the shaft lining concrete caused by temperature difference...owing to seasonal temperature changes...., the author did not clearly explain how to improve it. The paper will benefit greatly if the authors will add some discussion on how that aspect of sensor performance can be improved.

10.  Some future directions can be discussed in the conclusion part.

Reviewer 3 Report

the paper titled "Application of FBG sensor to safety monitoringof mineshaft lining structure " is very interesting and well structured. each section is clear and overall presents a useful study for the scientific community. the data acquired are elaborated very well and section 4 is complete. the conclusions and bibliography are appropriate to the article and the journal.

I only suggest that a better comparison be included in section 3 by highlighting the measurement uncertainty between the data acquired by strain gauge and the FBG strain sensor data. redo the graphs in figure 7 and 8 by inserting error bars.

the authors could enhance the introduction with this review article done on monitoring systems: "Internet of Things (IoT) for masonry structural health monitoring (SHM): Overview and examples of innovative systems" DOI:10.1016/j.conbuildmat.2021.123092.

Reviewer 4 Report

This paper calls: "Application of FBG sensor to safety monitoring of mine shaft lining structure" and concerned of underground mine shaft monitoring based on FBGs. The advantages of paper are interesting experimental results and good references list.

Meanwhile it’s not clear how to authors split impact of wavelength change (eq. 2) from temperature and from mechanical stress. Does temperature is constant or changing in small limits? Authors must be give answer on question of temperature impact.
